# Supplementation with Folic Acid and Cardiovascular Outcomes in End-Stage Kidney Disease: A Multi-Institution Cohort Study

**DOI:** 10.3390/nu14194162

**Published:** 2022-10-07

**Authors:** Yi-Ran Tu, Kun-Hua Tu, Cheng-Chia Lee, Pei-Chun Fan, Chieh-Li Yen, Victor Chien-Chia Wu, Ji-Tseng Fang, Yung-Chang Chen, Pao-Hsien Chu, Chih-Hsiang Chang

**Affiliations:** 1Kidney Research Center, Department of Nephrology, Chang Gung Memorial Hospital, Linkou Branch, Taoyuan 33305, Taiwan; 2Graduate Institute, Colleg of Medicine, Chang Gung University, Taoyuan 33305, Taiwan; 3Department of Cardiology, Chang Gung Memorial Hospital, Taoyuan 33305, Taiwan

**Keywords:** folic acid, cardiovascular outcome, end-stage kidney disease (ESKD)

## Abstract

Background: Folate is a water-soluble vitamin and is essential for maintaining cell functions. Dialysis removes folate, and folate deficiency is reported in patients with end-stage kidney disease (ESKD). However, there is no consensus as to the appropriate dosage of folate supplements and their advantages and disadvantages for patients with ESKD. Methods: This study was based on the electronic medical records of the Chang Gung Research Database (CGRD) of the Chang Gung Medical Foundation. We included patients who were diagnosed with ESKD, initiated hemodialysis, and were given folic acid supplements at any point from 1 January 2001 to 31 December 2019. The patients were divided into weekly and daily folic acid supplementation groups. We reduced the effects of confounding through the inverse probability of treatment weighting based on the propensity score. Results: We identified 2081 and 954 newly diagnosed patients with ESKD, who received daily and weekly folic acid supplements. The mean follow-up time was 5.8 years, and the event rates of arteriovenous access thrombosis were 17.0% and 23.6% in the daily and weekly folic acid supplementation groups (sub-distribution hazard ratio = 0.69, 95% confidence interval = 0.61 to 0.77), respectively. Neither group significantly differed in the occurrence of other clinical events, such as major cardiovascular cardiac events (e.g., myocardial infarction and ischemic stroke), all-cause mortality, cardiovascular death, infection death, malignancy, and adverse effects. Conclusion: a daily 5 mg folic acid supplementation might result in a lower event rate of arteriovenous access thrombosis in patients with ESKD than weekly folic acid supplementation. Further prospective studies are warranted to explore the preventive effect of folate on thrombosis.

## 1. Introduction

Folate, also known as vitamin B9, is essential for DNA and RNA synthesis and metabolizes amino acids, including homocysteine [1,2,3]. Patients with end-stage kidney disease (ESKD) are prone to nutritional deficiencies due to medication interactions, dietary restrictions, and malnutrition [1,4]. Furthermore, dialysis might remove folate by diffusion and lead to a folate deficiency after several weeks of therapy [1,4]. Additionally, folic acid supplements can improve all-cause mortality in patients with ESKD [1,5] and help to prevent cardiovascular disease [1,6]. Therefore, folic acid supplementation is recommended for patients with ESKD.

However, due to improvements in nutritional science over the last decade [7], the folic acid dosage and adverse effects should be re-evaluated for a greater precision and more tailored treatment of patients with ESKD. Therefore, we investigated the potential benefits of folic acid supplementation using a multicenter database.

## 2. Methods

### 2.1. Data Source

This study was based on the electronic medical records of the Chang Gung Research Database (CGRD) of the Chang Gung Medical Foundation. The database encompasses the data of the nationwide Chang Gung Memorial Hospital system, which is the largest health care system of its type in Taiwan, comprising two medical centers, two regional hospitals, and three district hospitals. The CGRD contains clinical epidemiological data, laboratory data, inpatient and outpatient records, emergency medical records, pathology reports, and disease category data. The overall coverage rate of the CGRD for the entire Taiwanese population is approximately 20% for outpatients and 12% for inpatients. More detailed information about the CGRD is available in other studies [8,9]. The disease diagnoses are coded using the International Classification of Diseases, Ninth Revision, Clinical Modification (ICD-9-CM) for records before 2016 and International Classification of Diseases, Tenth Revision, Clinical Modification (ICD-10-CM) for those after that year. This study was approved by the Institutional Review Board of the Chang Gung Memorial Hospital (approval number: CGMHIRB No. 2207290071).

### 2.2. Study Population and Patient Selection

We identified 11,766 patients diagnosed with ESKD, who initiated hemodialysis and took folic acid supplements between 1 January 2001 and 31 December 2019. The date of the initiation of hemodialysis was defined as the index date. Patients were excluded if they were younger than 20 years old before the index date; had underlying diseases including malignancy, renal transplantation, and autoimmune diseases; engaged in drug or alcohol abuse before the index date; or were pregnant on the index date. Additionally, we also excluded patients from the cohort if their follow-up duration was less than 90 days. Finally, we excluded patients who did not receive long-term dialysis in our hospitals. Long-term dialysis was defined as hemodialysis performed at least 8 times monthly during the first 3 months of follow-up. The final cohort enrolled in the study comprised of 2081 and 954 patients, who took folic acid supplements daily and weekly, respectively (Figure 1).

### 2.3. Folic Acid Usage

We are becoming increasingly aware of the beneficial effects of folic acid supplementation on the mortality risk of patients with ESKD [1,5,6,10]. However, the associations between different dosages of folic acid supplementation and patient outcomes are essentially unknown. Because or institution used a single form of folic acid (5 mg/tab), our low-dosage and high-dosage group comprised patients who took folic acid supplements weekly (1 tablet per week) and daily (1 tablet per day), respectively.

### 2.4. Covariates

The patient covariates included demographic information (age, sex, and body mass index), medical resource utilization (number of outpatient department visits and hospitalization in the previous year), primary renal diseases, comorbidities (hypertension, diabetes mellitus, and 13 others), Charlson’s comorbidity index score, and baseline laboratory and medication data. Primary renal diseases included interstitial nephritis, obstructive nephritis, polycystic kidney disease, hypertension nephropathy, diabetes nephropathy, chronic glomerulonephritis, and others (e.g., lupus). Baseline laboratory data included hemoglobin, the platelet count, albumin, blood urea nitrogen, creatinine, potassium, and sodium. Baseline medications included angiotensin-converting enzyme inhibitor/angiotensin receptor blocker, loop diuretics, and 13 others. The comorbidities were detected using ICD diagnostic codes in the case of at least 2 outpatient diagnoses or an inpatient diagnosis being confirmed before the index date.

### 2.5. Outcomes

Outcomes consisted of clinical events and laboratory outcomes. Clinical events were classified as primary mortality outcomes, secondary outcomes, and adverse-effect-related outcomes. The dates and causes of mortality were obtained in reference to the Taiwan Death Registry database. We could therefore discern cardiovascular deaths and infection-induced deaths. Secondary outcomes were sepsis, a major cardiovascular cardiac event (including acute myocardial infarction, acute ischemic stroke, and cardiovascular death), an acute myocardial infarction, an acute ischemic stroke, heart failure hospitalization, an intracranial hemorrhage, a new diagnosis of peripheral vascular disease, arteriovenous access thrombosis (fistula or graft), and a new diagnosis of malignancy. Peripheral vascular disease and malignancy were indicated by at least 2 outpatient diagnoses or 1 inpatient diagnosis. The occurrence of other secondary outcomes was indicated by an inpatient diagnosis. The adverse-effect-related outcomes were adverse effects on the gastrointestinal system (e.g., from orally administered proton pump inhibitors, metoclopramide, mosapride, or domperidone) and insomnia (e.g., from benzodiazepine or zolpidem). For each clinical event, the patients’ data were analyzed until the occurrence of the clinical event, the date of death, or the last hospital visit.

The laboratory outcomes of interest were changes in the hemoglobin, potassium, intact parathyroid hormone (iPTH), albumin, and high sensitivity c-reactivity protein (hsCRP) from baseline levels up to the 12th month after the index date. The laboratory outcome analysis was restricted to patients with baseline and 12-month follow up data for each laboratory parameter.

### 2.6. Statistical Analysis

The baseline characteristics of the patients taking daily versus weekly folic acid supplements were balanced through the inverse probability of treatment weighting (IPTW) based on the propensity score. The propensity score was defined as the predicted probability of being in the daily group in light of certain covariate values and was derived from a multivariable logistic regression model. All covariates (listed in Table 1), except for the follow-up year, were included in the propensity score calculation. We also included the index date as a covariate when calculating the propensity score so as to consider the secular trend. To prevent extreme weights from significantly affecting the results, we truncated the weight at the 99th percentile. Due to considerable missing laboratory data, the missing values were assigned using the single expectation–maximization imputation, and the IPTW of the data was calculated thereafter. The balance of the baseline characteristics between the 2 groups was assessed using the standardized difference (STD), where an absolute STD value of less than 0.1 was considered to indicate a negligible difference.

The risks of fatal outcomes (all-cause mortality, cardiovascular death, infection death and major cardiovascular cardiac event) between groups were compared using the Cox proportional hazard model. The incidence of non-fatal outcomes (e.g., arteriovenous access thrombosis ) between groups was compared using the Fine and Gray sub-distribution hazard model, which considers all-cause mortality a competing risk. The study groups (daily vs. weekly) were the only explanatory variables in the survival analyses. The changes in the laboratory data from baseline to the 12th month of follow-up were compared between groups using the generalized estimating equation (GEE). The GEE model included the intercept, the main effects of the study groups and time, and an interaction term. The difference in the laboratory data between groups was deemed notable when the interaction effect was significant.

A two-sided *p* value of less than 0.05 was considered statistically significant. All analyses were conducted with SAS software, Version 9.4 (SAS Institute, Cary, NC, USA).

## 3. Results

### 3.1. Basic Characteristics of the Study Patients

During the inclusion period, we identified 11,766 patients with ESKD, who initiated hemodialysis combined with folic acid usage. When the exclusion criteria were applied, 3035 patients remained in the sample for the analysis. Among these patients, 2081 patients received daily folic acid supplementation and the other 954 patients received weekly folic acid supplementation (Figure 1).

The mean age was 61.5 years (SD = 13.8 years), and men were more predominant (*n* = 1672, 55%). Generally, the baseline characteristics between the two groups did not differ substantially before the IPTW adjustment. Patients with daily folic acid supplement usage had a higher rate of hospitalization in the previous year (68.6% vs. 63.5%, STD = 0.11), greater platelet count (STD = 0.11), lower levels of blood urea nitrogen (STD = −0.15) and serum creatinine (STD = −0.14). They received less vitamin D therapy supplementation (STD = −0.12) and more vitamin B therapy supplementation (STD = 0.27) and showed a higher rate of steroid uses (STD = 0.11) than patients with weekly folic acid supplement usage. After the IPTW adjustment, the difference in the baseline characteristics between the groups was negligible, with all absolute STD values of less than 0.1 (Table 1).

### 3.2. Clinical Events

The mean follow-up period was 5.8 years (SD = 4.2 years). No significant difference in the morality outcomes between groups was noted (Table 2). The results showed that the risk of arteriovenous access thrombosis (fistula or graft) was significantly lower in patients with daily supplement use than those with weekly supplement use (17% vs. 23.6%; sub-distribution hazard ratio 0.69, 95% confidence interval 0.61 to 0.77). The risk difference was observed during a short follow-up period and was amplified afterward until approximately the 4th year of follow-up (Figure 2). Additionally, no significant difference in the remaining clinical events was observed, including the adverse-effect-related outcomes (Table 2).

### 3.3. Laboratory Outcomes

The changes in the evaluated laboratory data from baseline to the 12th month of follow-up for both groups are listed in Table 3. The hemoglobin, potassium, iPTH, and albumin levels showed no significant difference between the groups (*p* for interaction > 0.05). The level of hsCRP did not significantly increase from baseline to the 12th month in the daily supplement group (7.5 vs. 7.8 mg/dL); however, it significantly increased in the weekly supplement group (6.5 vs. 9.0 mg/dL, *p* < 0.05). Noticeably, the level of the increase was significantly larger in the weekly supplement group than in the daily supplement group (*p* for interaction = 0.027; Figure 3).

## 4. Discussion

Folate is an essential cofactor in one-carbon metabolism and suppresses hyperhomocysteinemia by promoting the remethylation of homocysteine. Homocysteine is an intermediary amino acid formed by the conversion of methionine into cysteine and has been demonstrated to induce vascular injuries, increase smooth muscle cell proliferation, enhance collagen production, promote leukocyte recruitment, decrease endothelial antithrombotic activity, and impair DNA methylation [5,11,12,13,14,15]. Furthermore, increased homocysteine is associated with an increased number of cardiovascular events in the standard population and patients with renal failure [16]. Folic acid supplements can lower the serum homocysteine levels [17,18], improve endothelial dysfunction [19,20], and reduce the rate of restenosis and the likelihood of cardiovascular events after a percutaneous coronary intervention [21,22]. Patients with renal dysfunction are more likely to have a downregulated methionine cycle, leading to lowered homocysteine levels [5,23].

Although folate supplements suppress homocysteine in patients with ESKD undergoing dialysis, their benefits in providing cardiovascular protection remain controversial [1,5,24].

Moreover, researchers remain uncertain regarding the appropriate daily amount of supplemented folic acid for ESKD. A lower serum folic acid level is associated with a 15% higher risk of all-cause mortality after the adjustment for case-mix covariates; however, the odds ratio was not significant after a further adjustment for laboratory variables [1]. Chien et al. found that a > 5 mg per day dose of folic acid had more beneficial effects than a ≤ 5 mg per day dose [5]. A meta-analysis showed no significant difference between lower doses (≤5 mg/day) and higher dose of folic acid (>5 mg/day) in their effects on the primary cardiovascular outcomes of patients with ESKD [6].

In our study, the rates of mortality and cardiovascular events in those receiving 5 mg per day of folic acid and those receiving 5 mg per week showed no significant difference. However, we found fewer occurrences of arteriovenous shunt thrombosis in the group taking 5 mg of folic acid daily. This implies that a larger folic acid supplement could prevent thrombosis events in patients with ESKD. A similar study also found folic acid supplements to be associated with lower adverse effects in patients with deep vein thrombosis [25]. Although we had no access to folate and homocysteine data, the hsCRP levels were lower among patients who took 5 mg of folic acid supplements daily rather than 5 mg weekly. Folic acid supplements might suppress homocysteine-triggered inflammation [26]. Further randomized controlled clinical trials are warranted to explore the preventative effect of folic acid supplements on thrombosis in patients with ESKD.

The National Institute of Health recommends that adults limit their intake of folic acid from fortified foods and supplements to 1 mg daily. Therefore, we must consider high-dose supplements’ potential clinical adverse effects (5 mg/day). Short-term adverse effects, such as gastrointestinal upset and insomnia, did not vary between the two different dosages. Our research found no substantial difference between the two groups in terms of long-term adverse effects, such as cancer. In our study, patients who took folic acid supplements daily had increased medical costs, but their financial status did not significantly differ with that of their counterparts taking supplements weekly at the end of follow-up. Therefore, if the pill burden is not a concern, folic acid supplements should be taken daily rather than weekly by patients with ESKD.

Our study has several limitations. Firstly, we had no data on the serum levels of folate and homocysteine. Although studies have recommended lowering the homocysteine levels with folate supplements, we could not identify patients with a folate deficiency in our dialysis population. Therefore, further applications of our results should consider the individual’s nutritional status. Secondly, our study focused solely on Taiwanese citizens, for whom rice is a staple in their diet. Future studies must consider diet-related differences in natural folic acid intake between cultures. Thirdly, vascular access thrombosis or deep vein thrombosis are multifactorial conditions associated with a different etiology, pathophysiology, and risk factors. All outcomes, except in the case of vascular access thrombosis, showed no difference between the daily and weekly doses; therefore, we could not exclude the possibility of change. Future studies could further investigate and analyze the association between folic acid and vascular thrombosis in dialytic patients. Finally, our study was limited by the fact we only analyzed one supplement dosage (5 mg). Future research should investigate outcomes for dosages under 5 mg.

## 5. Conclusions

Patients with ESKD had a lower event rate of arteriovenous access thrombosis and lower hsCRP levels when they took a 5 mg folic acid supplement daily rather than weekly. The all-cause mortality, CV mortality, infection death, other secondary outcomes, and drug-related adverse effects were not significantly associated with the frequency (daily or weekly) of supplement intake. Further studies should explore the relationship between the dosage and thromboembolic effects.

## Figures and Tables

**Figure 1 nutrients-14-04162-f001:**
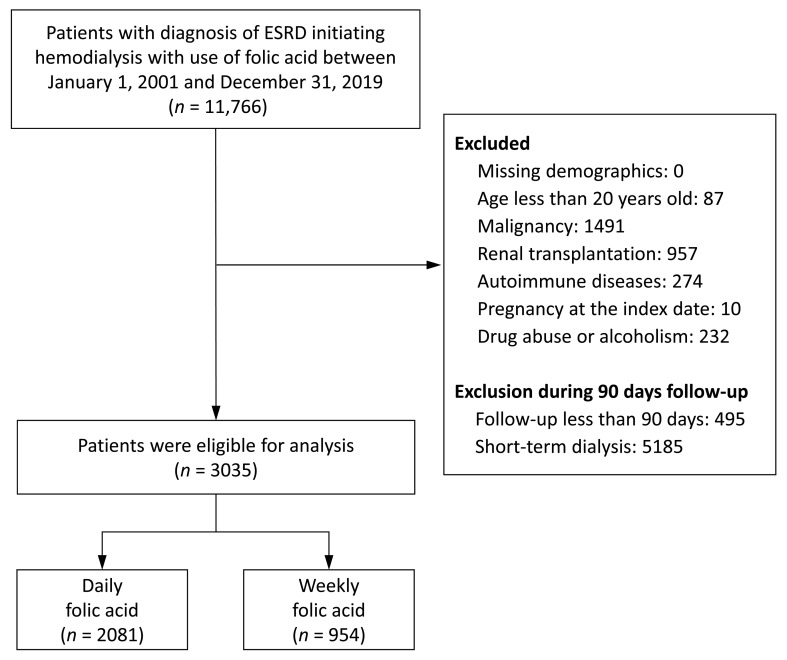
The illustration of the inclusion and exclusion criteria of the study patients. ESRD, end-stage renal disease.

**Figure 2 nutrients-14-04162-f002:**
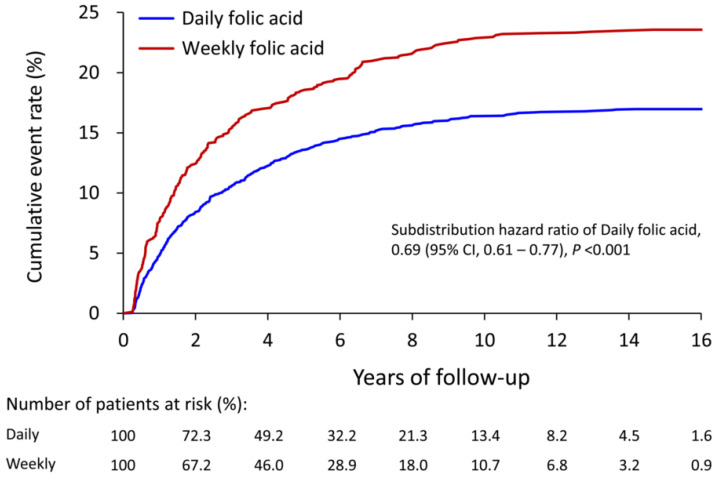
The cumulative event rate for patients receiving daily versus weekly usage of folic acid in the IPTW-adjusted cohort. IPTW, inverse probability of treatment weighting.

**Figure 3 nutrients-14-04162-f003:**
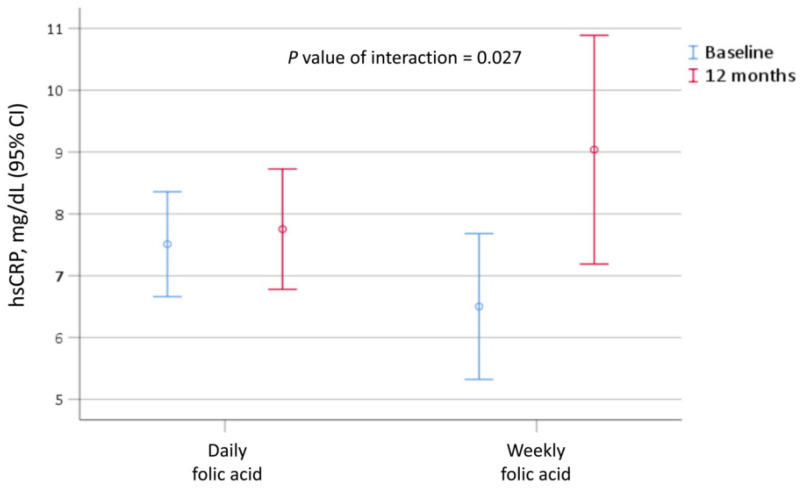
The level of hsCRP upon the initiation of hemodialysis and the 12th month of follow-up for patients receiving daily versus weekly folic acid in the original cohort. hsCRP, high sensitivity c-reactive protein.

**Table 1 nutrients-14-04162-t001:** Baseline characteristics of the patients.

Variable	Available Number	Before Imputation and IPTW			After Imputation and IPTW		
		Daily Folic Acid (*n* = 2081)	Weekly Folic Acid (*n* = 954)	STD	Daily Folic Acid	Weekly Folic Acid	STD
Demographics							
Age, years	3035	61.8 ± 14.1	61.1 ± 13.1	0.06	61.6 ± 13.8	61.6 ± 13.7	<0.01
Male, *n* (%)	3035	1143 (54.9)	529 (55.5)	−0.01	55.1%	55.4%	−0.01
Body mass index, kg/m^2^	1953	25.0 ± 5.1	25.4 ± 5.2	−0.09	24.8 ± 4.5	24.8 ± 4.1	<0.01
Number of OPD visits in the previous year	3035						
0		139 (6.7)	65 (6.8)	−0.01	6.7%	6.7%	<0.01
1–5		443 (21.3)	190 (19.9)	0.03	20.9%	21.2%	−0.01
6–15		634 (30.5)	299 (31.3)	−0.02	30.7%	30.5%	<0.01
≥16		865 (41.6)	400 (41.9)	−0.01	41.8%	41.6%	<0.01
Hospitalization in the previous year	3035	1428 (68.6)	606 (63.5)	0.11	66.9%	66.5%	0.01
Primary renal disease	3035						
Interstitial nephritis		15 (0.7)	8 (0.8)	−0.01	0.7%	0.8%	−0.01
Obstructive nephritis		457 (22.0)	163 (17.1)	0.12	20.4%	20.3%	<0.01
Polycystic kidney disease		30 (1.4)	18 (1.9)	−0.03	1.5%	2.0%	−0.03
Hypertension nephropathy		798 (38.3)	352 (36.9)	0.03	38.0%	38.6%	−0.01
Diabetes nephropathy		218 (10.5)	119 (12.5)	−0.06	10.9%	11.1%	<0.01
Chronic glomerulonephritis		492 (23.6)	253 (26.5)	−0.07	24.8%	23.6%	0.03
Others (e.g., lupus)		71 (3.4)	41 (4.3)	−0.05	3.6%	3.8%	−0.01
Charlson’s comorbidity index score	3035	4.4 ± 2.3	4.3 ± 2.2	0.05	4.4 ± 2.3	4.4 ± 2.2	<0.01
Comorbidity							
Hypertension	3035	1753 (84.2)	786 (82.4)	0.05	83.8%	83.7%	<0.01
Diabetes mellitus	3035	1162 (55.8)	519 (54.4)	0.03	55.3%	54.8%	0.01
Dyslipidemia	3035	873 (42.0)	380 (39.8)	0.04	41.3%	40.7%	0.01
Coronary artery disease	3035	409 (19.7)	177 (18.6)	0.03	19.4%	19.2%	0.01
Myocardial ischemia	3035	103 (4.9)	43 (4.5)	0.02	4.8%	4.7%	0.01
Ischemic stroke	3035	156 (7.5)	63 (6.6)	0.03	7.3%	7.5%	−0.01
Hemorrhagic stroke	3035	36 (1.7)	13 (1.4)	0.03	1.6%	1.6%	<0.01
Peripheral vascular disease	3035	121 (5.8)	44 (4.6)	0.05	5.4%	4.8%	0.02
Venous thrombosis	3035	31 (1.5)	8 (0.8)	0.06	1.3%	1.3%	<0.01
Atrial fibrillation	3035	75 (3.6)	35 (3.7)	<0.01	3.5%	3.5%	0.01
Heart failure hospitalization	3035	262 (12.6)	125 (13.1)	−0.02	12.6%	11.8%	0.02
Chronic obstructive pulmonary disease	3035	201 (9.7)	89 (9.3)	0.01	9.7%	9.9%	−0.01
Liver cirrhosis	3035	63 (3.0)	28 (2.9)	0.01	3.0%	3.1%	<0.01
Peptic ulcer disease	3035	363 (17.4)	180 (18.9)	−0.04	17.9%	18.2%	−0.01
Gouty arthritis	3035	478 (23.0)	198 (20.8)	0.05	22.5%	22.3%	<0.01
Laboratory data at baseline							
Hemoglobin, g/dL	2733	8.3 ± 1.2	8.3 ± 1.3	0.04	8.3 ± 1.2	8.3 ± 1.2	<0.01
Platelet count, ×10^3^	2694	208.0 ± 70.7	199.9 ± 74.5	0.11	205.6 ± 66.4	207.0 ± 72.8	−0.02
Albumin, g/dL	2731	3.9 ± 0.4	3.9 ± 0.4	−0.04	3.9 ± 0.4	3.9 ± 0.4	−0.01
Blood urea nitrogen, mg/dL	2734	90.7 ± 30.6	95.5 ± 33.0	−0.15	92.1 ± 29.9	91.8 ± 29.8	0.01
Creatinine, mg/dL	2734	10.2 ± 3.3	10.7 ± 3.4	−0.14	10.4 ± 3.3	10.4 ± 3.3	−0.01
Potassium, mEq/L	2736	5.1 ± 0.8	5.2 ± 0.8	−0.07	5.1 ± 0.8	5.1 ± 0.8	−0.01
Sodium, mEq/L	2673	138.4 ± 3.4	138.4 ± 3.5	−0.01	138.4 ± 3.2	138.4 ± 3.3	−0.01
Medication at baseline							
Antiplatelet	3035	713 (34.3)	303 (31.8)	0.05	33.6%	33.9%	−0.01
Anticoagulation	3035	34 (1.6)	14 (1.5)	0.01	1.6%	1.5%	0.01
ACEi/ARB	3035	1234 (59.3)	576 (60.4)	−0.02	59.9%	60.1%	<0.01
Beta blockers	3035	675 (32.4)	277 (29.0)	0.07	31.4%	31.1%	0.01
Calcium channel blocker	3035	1562 (75.1)	678 (71.1)	0.09	74.0%	74.2%	<0.01
Loops diuretics	3035	1429 (68.7)	652 (68.3)	0.01	68.7%	68.5%	<0.01
Nitrates	3035	653 (31.4)	298 (31.2)	<0.01	31.3%	30.6%	0.02
Vasodilator	3035	462 (22.2)	200 (21.0)	0.03	21.8%	21.4%	0.01
Oral hypoglycemic agents	3035	813 (39.1)	346 (36.3)	0.06	38.2%	38.4%	<0.01
Insulin	3035	736 (35.4)	331 (34.7)	0.01	35.0%	34.4%	0.01
Statin	3035	691 (33.2)	301 (31.6)	0.04	32.7%	32.1%	0.01
Vitamin D therapy	3035	134 (6.4)	93 (9.7)	−0.12	7.5%	7.4%	0.01
Vitamin B therapy	3035	1490 (71.6)	563 (59.0)	0.27	67.7%	68.1%	−0.01
Proton pump inhibitor	3035	545 (26.2)	226 (23.7)	0.06	25.6%	26.0%	−0.01
Steroid	3035	463 (22.2)	172 (18.0)	0.11	20.9%	20.5%	0.01
Follow-up year	3035	5.7 ± 4.2	5.9 ± 4.3	−0.06	5.8 ± 4.2	5.7 ± 4.2	0.01

Abbreviations: IPTW, inverse probability of treatment weighting; STD, standardized difference; OPD, outpatient department; ACEi, angiotensin-converting enzyme inhibitor; ARB, angiotensin receptor blocker. Data are presented as mean ± standard deviation or number (percentage).

**Table 2 nutrients-14-04162-t002:** Clinical events of patients with daily versus weekly use of folic acid in the IPTW-adjusted cohort.

Variable	Daily	Weekly	HR or SHR (95% CI) for Daily	*p* Value
Primary outcome				
All-cause mortality	49.2%	51.7%	0.95 (0.85–1.06)	0.335
Cardiovascular death	30.9%	30.4%	1.01 (0.87–1.17)	0.893
Infection death	23.3%	23.9%	0.97 (0.82–1.14)	0.673
Secondary outcome				
Sepsis	44.7%	44.4%	1.01 (0.94–1.09)	0.743
Major cardiovascular cardiac event *	42.4%	42.1%	0.99 (0.87–1.12)	0.836
Acute myocardial infarction	13.9%	14.1%	0.98 (0.86–1.12)	0.731
Acute ischemic stroke	13.1%	14.2%	0.92 (0.80–1.05)	0.215
Heart failure hospitalization	22.1%	22.3%	0.99 (0.89–1.10)	0.837
Intracranial hemorrhage	3.9%	3.9%	0.99 (0.77–1.28)	0.961
New diagnosis of peripheral vascular disease	11.9%	12.2%	0.98 (0.84–1.13)	0.729
Arteriovenous access thrombosis (fistula or graft)	17.0%	23.6%	0.69 (0.61–0.77)	<0.001
New diagnosis of malignancy	11.1%	11.3%	0.98 (0.84–1.14)	0.773
Side-effect-related outcome				
Gastrointestinal side effects of drugs	71.0%	69.1%	1.04 (0.98–1.11)	0.162
Use of oral form proton pump inhibitors	62.0%	60.3%	1.05 (0.98–1.12)	0.150
Insomnia with use of relevant drugs	21.5%	22.2%	0.96 (0.86–1.06)	0.399

Abbreviations: IPTW, inverse probability of treatment weighting; HR, hazard ratio; SHR, sub-distribution hazard ratio; CI, confidence interval. * Acute myocardial infarction, acute ischemic stroke, and/or cardiovascular death. Data are presented as percentages.

**Table 3 nutrients-14-04162-t003:** Laboratory examinations of interest at baseline and 12-month follow-up in the original cohort #.

Variable	Daily Folic Acid			Weekly Folic Acid			*p* for
	Baseline	12th Month	Change	Baseline	12th Month	Change	Interaction
Hemoglobin, g/dL	8.3 ± 1.3	10.3 ± 1.2	1.95 ± 1.55 *	8.3 ± 1.3	10.3 ± 1.4	2.07 ± 1.66 *	0.103
Potassium, mEq/L	5.1 ± 0.8	4.7 ± 0.8	−0.42 ± 0.94 *	5.2 ± 0.9	4.7 ± 0.8	−0.46 ± 1.01 *	0.339
iPTH, pg/mL	325.8 ± 317.0	221.7 ± 250.8	−104.1 ± 286.7 *	356.5 ± 334.2	263.7 ± 319.3	−92.8 ± 302.0 *	0.511
Albumin, g/dL	3.85 ± 0.41	3.86 ± 0.47	0.01 ± 0.45	3.87 ± 0.39	3.84 ± 0.49	−0.02 ± 0.44	0.083
hsCRP, mg/dL	7.5 ± 12.0	7.8 ± 13.8	0.24 ± 15.89	6.5 ± 10.9	9.0 ± 17.1	2.54 ± 18.21 *	0.027

Abbreviations: iPTH, intact parathyroid hormone; hsCRP, high sensitivity c-reactive protein. # The analysis was restricted to the patients with available data both at baseline and 12-month follow-up for each laboratory parameter; * indicates that the change from baseline to 12th month follow-up was significant.

## Data Availability

The data presented in this study are available on request from the corresponding author. The data are not publicly available due to the access of Chang Gung Research Database is required the permission of Chang Gung Memorial hospital system.

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
