# Peer review of "Supplementation with Folic Acid and Cardiovascular Outcomes in End-Stage Kidney Disease: A Multi-Institution Cohort Study"

_nutrients, 2022, doi:10.3390/nu14194162_

Round 1
Reviewer 1 Report
This is a fine paper.
Several points:
Line 68 - do not most renal conditions result from "underlying conditions"? e.g., diabetes, lupus, IgA nephropathy, PCKD, etc. Please explain.
Lines 87, 88 - are not many of the conditions also part of the Charlson index?
Line 217 - no is not
Conclusion - all your outcomes show no difference between daily vs weekly FA use. Perhaps your thrombosis outcome arose by chance? Please consider this possibility.
Author Response
1. Line 68 - do not most renal conditions result from "underlying conditions"? e.g., diabetes, lupus, IgA nephropathy, PCKD, etc. Please explain.
Reply: Thank you for your comment. The renal conditions about primary renal diseases in our study were interstitial nephritis, obstructive nephritis, polycystic kidney disease, hypertensive nephropathy, diabetic nephropathy, chronic glomerulonephritis, and others. We did not present these data in the original manuscript but in fact we did consider this in the statistical analysis. We have revised the content in the revised manuscript in the paragraph of “Table 1” and the Covariates subsection of the Methods section.
2. Lines 87, 88 - are not many of the conditions also part of the Charlson index?
Reply: We truly appreciate your comment. The patient covariates about comorbidities in our research were part of the Charlson Comorbidity Index (CCI), such as diabetes, stroke, etc.
3. Line 217 - no is not
Reply: We truly appreciate your comment. We have revised the content in the revised manuscript.
4. Conclusion - all your outcomes show no difference between daily vs weekly FA use. Perhaps your thrombosis outcome arose by chance? Please consider this possibility.
Reply: We truly appreciate your comment. We have revised the content in the third point of the limitations of the “Discussion” section in the revised manuscript.

Reviewer 2 Report
In this multicenter retrospective study (n=3035) using database from Taiwan, the author compared the outcomes in patients with ESRD who were taking folic acid supplementation 5 mg daily vs. 5 mg weekly. The analysis found a significant reduction in vascular access thrombosis in those on daily folic acid supplementation. Over all this is a well performed analysis and written well. The study has clinical relevance.
There are multiple factors that can contribute to vascular access thrombosis in dialysis patients including proximal vein stenosis manifested as increase in venous pressure. A more detailed discussion on limitations should be included.
Author Response
There are multiple factors that can contribute to vascular access thrombosis in dialysis patients including proximal vein stenosis manifested as increase in venous pressure. A more detailed discussion on limitations should be included.
Reply: We truly appreciate your comment. Vascular access thrombosis is a multifactorial condition which associated with different etiology, pathophysiology, and risk factors. We have revised the content in the third point of the limitations of the “Discussion” section in the revised manuscript.
